# Ex-Vivo Treatment of Tumor Tissue Slices as a Predictive Preclinical Method to Evaluate Targeted Therapies for Patients with Renal Carcinoma

**DOI:** 10.3390/cancers12010232

**Published:** 2020-01-17

**Authors:** Caroline Roelants, Catherine Pillet, Quentin Franquet, Clément Sarrazin, Nicolas Peilleron, Sofia Giacosa, Laurent Guyon, Amina Fontanell, Gaëlle Fiard, Jean-Alexandre Long, Jean-Luc Descotes, Claude Cochet, Odile Filhol

**Affiliations:** 1Université Grenoble Alpes, Inserm, CEA, IRIG-Biology of Cancer and Infection, UMR_S 1036, F-38000 Grenoble, France; caroline.roelants@inovarion.com (C.R.); qfranquet@chu-grenoble.fr (Q.F.); csarrazin1@chu-grenoble.fr (C.S.); nicolas.peilleron@gmail.com (N.P.); sofiagiacosa@gmail.com (S.G.); laurent.guyon@cea.fr (L.G.); claude.cochet@cea.fr (C.C.); 2Inovarion, 75005 Paris, France; 3Université Grenoble Alpes, Inserm, CEA, IRIG-Biologie à Grande Echelle, UMR 1038, F-38000 Grenoble, France; catherine.pillet@cea.fr; 4Centre hospitalier universitaire Grenoble Alpes, CS 10217, 38043 Grenoble CEDEX 9, France; lafontanell@chu-grenoble.fr (A.F.); g.fiard@ucl.ac.uk (G.F.); JALong@chu-grenoble.fr (J.-A.L.); jldescotes@chu-grenoble.fr (J.-L.D.)

**Keywords:** drug sensitivity, immune infiltration, renal cancer, targeted therapy, tumor slice culture

## Abstract

Clear cell renal cell carcinoma (ccRCC) is the third type of urologic cancer. At time of diagnosis, 30% of cases are metastatic with no effect of chemotherapy or radiotherapy. Current targeted therapies lead to a high rate of relapse and resistance after a short-term response. Thus, a major hurdle in the development and use of new treatments for ccRCC is the lack of good pre-clinical models that can accurately predict the efficacy of new drugs and allow the stratification of patients into the correct treatment regime. Here, we describe different 3D cultures models of ccRCC, emphasizing the feasibility and the advantage of ex-vivo treatment of fresh, surgically resected human tumor slice cultures of ccRCC as a robust preclinical model for identifying patient response to specific therapeutics. Moreover, this model based on precision-cut tissue slices enables histopathology measurements as tumor architecture is retained, including the spatial relationship between the tumor and tumor-infiltrating lymphocytes and the stromal components. Our data suggest that acute treatment of tumor tissue slices could represent a benchmark of further exploration as a companion diagnostic tool in ccRCC treatment and a model to develop new therapeutic drugs.

## 1. Introduction

Clear cell renal cell carcinoma (ccRCC) is the most frequent subtype of kidney cancer representing above 3% of all cancers. At the time of diagnosis, 30% of cases are metastatic and are associated with a poor prognosis and without long-lasting effects of traditional oncologic treatment such as chemotherapy or radiotherapy [1]. With the advance of targeted therapies for RCC, several agents targeting angiogenesis and signal transduction pathways such as sunitinib, temsirolimus, and pazopanib have appeared and showed improved clinical benefit and survival in randomized prospective clinical trials. Yet, improvements are still required, as many of these current therapies are limited by acquired resistance mostly through activation of alternative pathways [2]. The tumor immune microenvironment of ccRCC is known to be highly immunosuppressive and immune infiltration of tumors is closely associated with clinical outcome. Recently, immune checkpoint inhibitors have demonstrated significant anti-tumor activity in the first-line treatment of intermediate to poor risk RCC patients, but these therapies are only effective for a small fraction of patients, and are associated with problems, such as side effects and high costs [3,4,5,6]. Thus, new treatment strategies are needed to improve efficacy in a broader patient population. In the last decade, efforts have primarily focused on establishing a framework for predictions of anticancer drug responses using in vitro tumor cell line models [7,8,9,10,11,12,13,14]. These techniques are limited by the cell dissociation that selects the more robust cells and the ones that can attach to the cell culture substratum [15,16,17]. Moreover, in these conditions, inadequate representation of the tumor heterogeneity and microenvironment interactions during a preclinical screen can result in inaccurate predictions of drug candidate effects.

Organoids derived from patient tumors have recently gained much interest as promising tools for several translational applications, such as high-throughput drug screens and personalized medicine [18,19,20]. Tumor organoids grown with undefined natural (e.g., Matrigel^®^) or synthetic extracellular matrix gels show improved resemblance to the original tumor compared to 2D cultured cancer cell lines. However, they do not model tumor–stromal interactions (cancer cells, immune, and endothelial cells) and the growth selection pressures applied during their generation have the potential to introduce bias [13,21]. Consequently, the prediction of treatment outcome extrapolated from organoids may not recapitulate each cancer patient tumor. Moreover, it is not clear whether the timescales are quick enough to affect patient care [22]. 

Another approach often considered more representative is the use of patient-derived xenograft (PDX) systems. However, the generation of PDX models exhibits a low engraftment rate, and the timescales and costs involved in this process are very significant [23]. Furthermore, the PDX deviates from the original tumor over time [24], and difference in pathophysiology between animal models and humans contributes to high failure rates of current small-molecule inhibitors in preclinical trials [25,26,27]. Thus, predicting successful anticancer therapy remains extremely challenging, largely due to extensive inter- and intratumor heterogeneity [28] and there remains a need for alternative, innovative models that allow the precise balance between manipulability and biological complexity.

To address these challenges, ex-vivo culture of intact tumor slices is potentially an extremely attractive system that has been already validated in various types of cancers [29,30,31,32,33,34,35,36,37,38].

This method has several advantages: (1) it can be rapidly established using only small samples of fresh tissue with a limited cost, (2) it preserves the tumor architecture and the spatial interaction between tumor and stroma, (3) testing of drug susceptibility can be combined with gene sequencing and immunohistochemistry analysis. To the best of our knowledge, tumor slice culture has never been validated in renal carcinoma.

In this study, we developed different biological cell-based systems like 3D tumor spheroids, mice orthotopic tumor xenografts, and patient-derived tumor slice cultures (PDTSC) for ex-vivo assessment of drug effects in renal carcinoma. As we recently showed, a combination of two inhibitors targeting both the PI3K and Src kinases impedes cell viability of renal carcinoma cells [39], we compared the efficacy of this combination to standard-of-care-drugs for RCC like sunitinib, pazopanib, and temsirolimus using 3D tumor spheroids and PDTSC methods. We show that PDTSC has the potential to be exploited for cancer cell sensitivity assessment to novel molecularly targeted therapies among patients with ccRCC, and to identify suitable candidates for drug combinations in a cost-effective and patient-friendly manner. We also demonstrate that PDTSC faithfully preserves the molecular landscape of the original renal carcinoma, retaining histopathology, including the stromal components and the immune cells that innately infiltrate the patient’s malignant epithelial cells, features that can be potentially useful to evaluate predictive biomarkers of treatment response and for patient stratification in prospective trials with immune checkpoint inhibitors.

## 2. Results

### 2.1. Evaluation of Drug Sensitivity on 786-0 Cell-Derived Spheroids

We first compared the induction of cell mortality in 786-O spheroids after their treatment with either a combination of GDC-0941 and saracatinib (GDC/SRC), two small-molecule inhibitors that target the PI3K and Src kinases respectively [39], or the currently clinically used inhibitors sunitinib, pazopanib, or temsirolimus at the indicated concentrations. Treated spheroids were recorded for 48 h using an Essen IncuCyte Zoom live-cell microscopy instrument (Figure 1A). Cell death induced by the different treatments at 6, 12, 24, and 48 h, was quantified through propidium iodide (PI) incorporation normalized by the surface of the spheroid. The results show that the drug effects on 786-O spheroids could be easily quantified using the Incucyte microscopy instrument (Figure 1B). Moreover, monitoring the size of the spheroids after 36 h of treatment showed that the GDC/SRC combination induces a significant reduction of the spheroid size (35%) while the effects of the other drugs were weaker compared to DMSO for which spheroid area declined by 15%, probably due to the maturation of the organoids that were under culture condition for five days (Figure 1C). During the last 12 h of treatment, cells in the spheroid center that was hypoxic, might have begun to die. Next, immunohistochemistry was performed on paraffin-embedded spheroids to visualize both the cellular architecture and the cell proliferation inside the 3D-spheroids. As shown in Figure 1D–F, both the integrity of the spheroids and the cell proliferation detected by PCNA labeling were affected by the GDC/SRC combination or Temsirolimus treatments confirming their effects on spheroids viability. Moreover, spheroid area measurements (Figure 1E) were consistent with the analysis of PI incorporation determined with the Incucyte microscope (Figure 1A). However, although promising, these data obtained with a 3D cancer cell line model suffered from inherent limitations due to inadequate representation of the heterogeneous architecture of human tumor and tumor–stromal interactions, which renders the interpretation on efficacy testing challenging. This is attested by the observation that among all the new molecules discovered for their action on cancer cell line models, only very few reached the FDA agreement. Therefore, implementation of physiologically relevant in- vitro models closer to patient-derived tumors is required.

### 2.2. Tissue Slice Cultures of Renal Tumors

We set out to determine whether an ex-vivo treatment protocol could be used as a means of determining ccRCC sensitivity to various cytotoxic agents. The PDTSC methodology has been previously used to evaluate the drug sensitivity of normal and tumor tissues [29,30,31,32,33,34,35,36,37,38]. Therefore, we set up an adaptation of this method outlined in Figure 2A, as an ex-vivo protocol to examine responses of ccRCC to different therapeutic agents. Cultures of slices, obtained either from 786-O-derived tumors generated in mouse xenografts or from human ccRCC surgical resection specimens, were prepared as detailed in the Methods section, and then subjected to a variety of tests. First, we noticed that over 96 h of culture, luminescence measurement of 786-O-luc cells in the tumor slice remained constant, attesting their viability during this time schedule (Figure 2B).

### 2.3. The Cytotoxic Effects of Drug Treatments Can Be Evaluated in Tissue-Slice Cultures

In order to evaluate the PDTSC approach, we first used the renal carcinoma mouse xenograft model. The tumors were extracted from the mice, directly processed into 300 μm slices, and treated for 48 h as described in Materials and Methods and indicated in Figure 2A. Cell viability evaluated by ethidium homodimer staining of treated tumor-slice cultures are illustrated in Figure 3A. Mortality quantified on five to seven images using ImageJ, was reported as “Cell death/DMSO” that represents the percentage of dead cells in the different groups divided by the percentage of dead cells in the DMSO-treated slices. The mortality rate showed a significant difference between DMSO and drugs alone or the GDC/SRC combination (*p* < 0.05) (Figure 3A,B). Immuno-histochemistry (IHC) analysis was used to determine whether a differential proliferative (PCNA) response to drug treatment could be detected. For this, paraffin-embedded sections were stained with a PCNA antibody and counter-colored with hematoxylin. We found that the GDC/SRC combination caused a significant decrease in PCNA staining, while temsirolimus, sunitinib, and pazopanib were less efficient (Figure 3C). Taken together, these results demonstrate that PDTSC allows for the rapid investigation of ccCRCC sensitivity to targeted therapies.

To further evaluate the potential of this approach, slices from surgical resections of human ccRCC tumors were analyzed using the same optimized protocol. In this study, we focused on patient tumors that were later on characterized as renal clear cell carcinoma by a board certified histo-pathologist at the Urology Department—University Hospital Center of Grenoble-Alpes. We note that our protocol did not interfere with the pathologist’s analysis. Warm ischemia was reduced to 15 min including tumor dissection and extraction during the surgery. Cold ischemia between extraction and the beginning of the culture was less than 2 h (including tumor sample dissection, transport and slicing). Two small pieces from two distinct regions (A and B) of each tumor were taken and processed in slices using a Vibratome^®^. Then, each tumor slice was cultured in the presence of the vehicle (0.2% DMSO) or the indicated therapeutic agents and assayed for cell viability after 48 h of drug treatment. Figure 4A shows that sample A disclosed high sensitivity to pazopanib or the GDC/SRC combination whereas sunitinib and temsirolimus were almost without effect. In contrast, sunitinib significantly compromised cell viability in sample B. Samples B-treated slices were further analyzed for their proliferation status after fixation and paraffin inclusion to assess functional response and cell viability. PCNA staining was detected in DMSO-treated slices (11.9%), whereas very few cells were stained in sunitinib-treated slices (0.2%). Moreover, a strong staining of cleaved-caspase-3 that reflects apoptotic cell death was observed in sunitinib-treated tumor slices (42.1%) but almost undetectable in DMSO-treated samples (4.2%) (Figure 4B).

These results highlight the intra-tumor heterogeneity of ccRCC, a property that has been well documented by extensive multi-regional whole-genome and -exome sequencing [40]. Collectively, these data demonstrated that PDTSC can be used to assess functional response and cell survival of human renal carcinoma specimens to drug treatments, reinforcing its value as a companion diagnostic tool in ccRCC treatment.

### 2.4. Acute Ex-Vivo Drug Treatments Identify Renal Tumor Subsets with Distinct Therapeutic Profiles

We compared the cell death rate of four different patient tumors upon the same panel of drug treatments (Figure 4C). Interestingly, this approach allowed for the identification of differential patient responses revealing sensitive and resistant tumors. For example, pazopanib was completely inactive on NB029, YL024 and MD034, whereas it was the most efficient on NM014. Temsirolimus was without effect on YL024.

Inactivation of the Von Hippel–Lindau (VHL) tumor suppressor gene has been shown to play an important role in the process of angiogenesis in RCC. As a component of an E3 ubiquitin ligase complex, the VHL protein targets the hypoxia-inducible transcription factors (HIF1α and HIF2α) for degradation. Loss of VHL function in ccRCC leads to the constitutive stabilization of these transcription factors, leading to a highly angiogenic environment [41] and HIF2α has recently emerged as a therapeutic target in ccRCC [42]. In line with this, we determined the protein expression level of VHL, HIF1α and HIF2α in human renal tumor slices (Figure 4D, left panels). Immuno-staining quantification shows that in slice NM014 where VHL expression was undetectable, HIF1α and mainly HIF2α were more abundant than in slice GD022 where VHL was present (Figure 4D, right panels).

### 2.5. Predictive Biomarkers in Renal Tumor Slice Cultures

The trafficking of immune cells in human cancers affects their immunobiology but also could have a major prognostic and predictive impact on the efficacy of the patient treatment. Indeed, renal cell carcinoma is an immunogenic tumor that characteristically harbors abundant infiltrating lymphocytes [43] and it has been shown that across renal tumors, there is a wide range of immune infiltrates [44]. Therefore, we tracked immune cells and their interaction with cancer cells within fixed slice cultures of different patient tumor samples (Figure 5). As an example, Figure 5A shows representative images of two non-treated tumor slices ML025 and DP027 in which the microvessel density labeled by CD34 staining was similar (Figure 5, right panel). Tumor slice DP027 was infiltrated with fewer cytotoxic CD8^+^ T cells than the tumor slice ML025. Interestingly, it has been suggested that in highly infiltrated ccRCC tumors, T-cell activation state is a key determinant of ccRCC prognosis and likely of immunotherapy response. Given the variety of mechanisms triggered by molecularly targeted agents in cancers and their late-stage clinical trials, the validation of drug sensitivity predictive models may be critical to identify the right drug for the right patients and help to understand determinants of responsiveness, wherein alternative treatments could potentially overcome resistance [45]. There is a recently growing body of literature describing PDTSC from different normal and tumor tissues [34,35,36,37,38]. However, to our knowledge, the present study is the first to demonstrate the potential use of this approach to evaluate renal cancer response to novel therapies while modeling the tumor immune microenvironment. An important benefit of the PDTSC strategy is that it provides a rapid and easy readout of the functional effects and drug responses that result from a complex array of molecular alterations among patients with ccRCC. PDTSC delivers a much faster timeline than PDX animal models, which require at least 6 to 7 weeks to become established versus 48 h for the PDTSC method. There is an important limitation inherent to PDTSC: the frequent intra-tumor heterogeneity may not be represented in individual slices from specific regions of a surgical resection specimen. However, this can be taken into account by a careful geographical collection of replicated tumor slices. In agreement with the key role played by the immune infiltrate in ccRCC, a phase 3 clinical trial (CheckMate214) showed benefits in terms of overall survival and objective response rate using an immunotherapy combination (ipilinumab plus nivolumab) versus sunitinib for intermediate and poor-risk patients with previously untreated advanced renal cell carcinoma [46].

In both tumor slices, we also detected a differential intra-tumor positive staining for the protein tyrosine phosphatase receptor CD45, one of the key players in the initiation of T cell receptor signaling [47]. CD45^+^ cells were abundant in DP027 tumor and localized in close proximity with microvessels and red blood cells. These cells were more intricately distributed in the ML025 slice than in DP027 reflecting a potential immune infiltration.

The tumor microenvironment deploys various immune escape mechanisms that neutralize CD8 T cell-mediated tumor rejection. One mechanism implies the aberrant expression of programmed death-ligand 1 (PD-L1) that targets the neutralization of activated CD8 T cells. PD-L1 has been reported in several human cancers including RCC [48]. This ligand is aberrantly expressed on the surface of both primary and metastatic RCC tumor cells [49] and several studies have described a positive correlation between PD-L1 expression, metastasis, and poor outcomes in ccRCC [50]. Consistent with this, we found that PD-L1 was strongly expressed in the ML025 tumor slice while undetectable in the DP027 tumor slice. Of note, ML025 slices were both positive for PD-L1 and cytotoxic CD8^+^ T cells. Interestingly, it has been suggested that metastatic melanoma that are both expressing PD-L1 and CD8^+^ T cells will likely respond to immunotherapy [51].

The LIM1 transcription factor which is essential for the development of human kidney is reactivated in nephroblastomas [52] and implicated in the metastatic spread of ccRCC [53]. While being undetectable in the DP027 slice, a strong intratumor LIM1 expression was observed in the ML025 slice. Altogether, these data support the contention that the PDTSC method allows for precise, short-term modeling of the stromal/immune microenvironment of renal tumors.

### 2.6. Prediction of Potential Correlations between Drug Sensitivity Responses and Tumor Immune Infiltration

To investigate whether there are links between drug-sensitivity and specific biomarkers previously analyzed in Figure 5A, we performed correlation analysis (Figure 5B). Eighteen human renal tumors (26 tumor specimens) that have been challenged for both drug-sensitivity and IHC- specific labeling were compared. Pairwise correlated variables were plotted in a graph of correlation matrix according to correlation coefficients indicated either by colored circles or numbers. This analysis highlights three subgroups/clusters of correlations between biomarkers only, drugs only and both drugs and biomarkers. In the first cluster, we visualized a strong positive correlation between CD8 and CD45 expressions (correlation coefficient, 0.54) and between HIF2α and PD-L1 (correlation coefficient, 0.5). These results are consistent with the literature as CD8 cytotoxic T cells are a subpopulation of CD45 positive leucocytes [54] and HIF2α as a transcription factor, binds to the PD-L1 promotor to induce its expression [55]. Inside the “drug” cluster, correlation are high (correlation coefficients > 0.35 except for the temsirolimus compared to the sunitinib situations). This result can be explained because the four treatments tested have similar action mechanisms (all are kinase inhibitors), however, they are not equal and more samples could help to find differences. Finally, the most informative cluster that compares biomarkers and drug treatments highlights two positive correlations between CD45 and temsirolimus (TEM) (correlation coefficient, 0.49) and PD-L1 and SUN (correlation coefficient, 0.22) and a negative correlation between GDC/SRC and HIF1α (correlation coefficient, −0.28). TEM has been demonstrated to have immune-modulating activity [56]. Obviously, the degree of correlation should be established by increasing the amount of tumor tissue samples for IHC analysis but even with this small cohort of tumor samples, potential valuable correlations dawned in this analysis and warrants further investigations.

## 3. Discussion

Given the variety of mechanisms triggered by molecularly targeted agents in cancers and their late-stage clinical trials, the validation of drug sensitivity predictive models may be critical to identify the right drug for the right patients and help to understand determinants of responsiveness, wherein alternative treatments could potentially overcome resistance [45]. There is a recently growing body of literature describing PDTSC from different normal and tumor tissues [29,30,31,32,33,34,35,36,37,38]. However, to our knowledge, the present study is the first to demonstrate the potential use of this approach to evaluate renal cancer response to novel therapies while modeling the tumor immune microenvironment. An important benefit of the PDTSC strategy is that it provides a rapid and facile readout of the functional effects and drug responses that result from a complex array of molecular alterations among patients with ccRCC. PDTSC delivers a much faster timeline than PDX animal models, which require at least 6 to 7 weeks to become established versus 48 h for the PDTSC method. There are several limitations inherent to PDTSC: (1) the frequent intra-tumor heterogeneity may not be represented in individual slices from specific regions of a surgical resection specimen. However, this can be taken into account by a careful geographical collection of replicated tumor slices; (2) fresh primary tissue may not be available when needed (in case of recurrent disease) and radical nephrectomy is not always performed on metastasized patients, whereas screening of these patients would be highly beneficial in the context of a predictive assay. The proof that the PDTSC can identify the best treatment need further investigations. In particular, as 30% of ccRCC becomes metastatic, one third of the patients will probably need specific treatments. In this context, the therapeutic profiling generated from the PDTSC may be informative after a retrospective clinical follow-up from patients who will develop metastasis; (3) PDTSC may be not relevant for some active drugs that are metabolites (e.g., for sunitinib).

## 4. Materials and Methods

### 4.1. Reagents, Drugs and Antibodies

Saracatinib (SRC) and GDC-0941(GDC) were obtained from LC Laboratories (Woburn, MA, USA). temsirolimus (TEM), pazopanib (PAZO), and sunitinib (SUN) were purchased from Selleck Chemicals (Houston, TX, USA), propidium iodide and Hoechst 33342 from Sigma-Aldrich (St Louis, MO, USA), and Live & Dead kit from Life Technologies (Carlsbad, CA, USA). The antibodies against the following targets were used: PCNA, CD8, CD34, PD-L1 (Ab29, Ab101500, Ab81289, Ab205921, Abcam, Cambridge, UK), Cleaved-Caspase-3, CD45 (#9664, #13917, Cell Signaling, Danvers, MA, USA); HIF1α, HIF2α (NB100-479, NB100-122, Novus Biologicals, Centennial, CO, USA), VHL (MA-1-12638, Thermo Scientific, Waltham, MA, USA).

### 4.2. 3D-Spheroid Culture and Live Cell Tracking

786-O cells (ATCC-CRL-1932) are derived from a human primary clear cell adenocarcinoma. This highly metastatic cell line is negative for VHL and is cultured in RPMI-1640 Medium supplemented with 10% SVF and penicillin [100 U/mL], streptomycin [100 µg/mL].

Spheroids were prepared in 96-wells U-bottom with low evaporation lid (MicrotestTM, Becton Dickinson Labware, San Jose, CA, USA) coated with 20 mg/mL poly-HEMA (Sigma-Aldrich). A 786-O cell suspension (1 × 10^3^ cells) was seeded in each well and cells were allowed to form spheroids within three days. Then, they were treated with indicated inhibitors for 48 h in the presence of Propidium iodide (0.5 µg/mL) to visualize dead cells and video recorded every hour using an Incucyte microscope, an automated live cell imager with high-throughput capabilities and built-in data analysis (Essen Biosciences, Welwyn Garden City, UK). Experiments were conducted at 37 °C and 5% CO_2_. Quantification of cell death was measured after 48 h as a percentage of confluence in the red channel (PI%) using the software incorporated into the IncuCyte Zoom. To normalize the data, all values for each time point was divided by the value at *T*_0_. Experimental data are shown as mean ± standard error mean (SEM) except for Figure 1E for which whole the points are shown overlaid on boxplots and whiskers. Classically, the box corresponds to the first and third quartiles, and the horizontal bar is the median, whereas the whiskers demarcate here the extreme values.

### 4.3. Mice Orthotopic Tumor Xenograft Models

All animal studies were approved by the institutional guidelines and those formulated by the European Community for the Use of Experimental Animals. Six week-old BALB/c female nude mice (Charles River Laboratories, Wilmington, MA, USA) with a mean body weight of 18–20 g were used to establish orthotopic xenograft tumor models. The mice were housed and fed under specific pathogen-free conditions. To produce tumors, renal cancer cells 786-O-luc (Roelants et al.) were harvested from sub-confluent cultures by a brief exposure to 0.25% trypsin-EDTA. Trypsinization was stopped with medium containing 10% FBS, and the cells were washed once in serum-free medium and resuspended in 500 µL PBS. Renal orthotopic implantation was carried out by injection of 3 × 10^6^ 786-O luc cells into the left kidney of athymic nude mice. Mice were weighed once a week to monitor their health and tumor growth was measured by imaging luminescence of 786-O-luc cells (IVIS).

### 4.4. Patients and Clinical Samples

All human renal carcinoma samples were obtained from patients, with their informed consent and all procedures were approved by the ethic committee (Patient protection committee No 2017 A0070251). All patients had serology to detect blood transmissible diseases before surgery and all samples were anonymized. Fresh renal tumor tissues were obtained from patient undergoing a partial or a total nephrectomy for cancer at the Urology Department—University Hospital Center of Grenoble-Alpes (CHUGA). The minimal size of tumor samples for inclusion was 2 cm. After resection, tissue samples were directly transported to the pathology department of the CHUGA in a cold saline solution (Sterile 0.9% NaCl). A macroscopic dissection was performed by a pathologist and as far as possible two distinct tumor samples (A and B) were placed in a sterile conical tube containing a conservation medium (ice-cold sterile balanced salt HBSS solution containing [100 U/mL] penicillin and [100 µg/mL] streptomycin) on wet ice during transport from the pathology department to the INSERM research laboratory (CEA).

### 4.5. Preparation of Tissue Slices and Organotypic Culture

Upon arrival, resections were manually minced using a sterile scalpel and samples were soaked in ice-cold sterile HBSS, orientated, mounted in low-melting agarose (5%), and immobilized using cyanoacrylate glue. Thick tissue slices (300 μm) were prepared from fresh tissue under sterile conditions using a Vibratome VT1200 (Leica Microsystems, Wetzlar, Germany). Slicing speed was optimized according to tissue density and type; in general, slower slicing speed was used on the softer tissues and vice versa (0.2–0.7 mm/s). Vibration amplitude was set between 1.85 to 2.45 mm.

Tissue slices were then carefully placed on 0.4 μm pore size Teflon membrane culture inserts (Millipore Corporation, Burlington, MA, USA) containing one slice per insert and cultured for up to 96 h. at 37 °C in a 5% CO_2_ humidified incubator using 2 mL of DMEM media supplemented with 20% inactivated FBS (GIBCO), 100 U/mL penicillin (Invitrogen, Carlsbad, CA, USA). Inserts were placed in a rotor agitator to allow gas and fluid exchanges with the medium. For each tumor, two slice samples (A and B) were treated with the inhibitors at the indicated concentrations for 48 h.

### 4.6. Slice Viability Assay

At the end of treatment, lived slices were stained with the Live & Dead kit (Life Technologies) as recommended and nuclei were labeled with Hoechst 33342. Images were taken with an Apotome-equipped Zeiss Axio-Imager microscope with a 20× PlanApochromat objective (Numerical Aperture 0.8). A minimum of three regions of interest (ROI) were taken, at three positions in z with 7 µm intervals to avoid counting the same nuclei twice. Dead cells in the tissue slices were quantified with scripts of ImageJ and further processed with R version 3.4.3 [57]. The histogram of red fluorescence intensity shows a peak of low intensity corresponding to live cells, and high and widely spread intensity values, corresponding to dying cells., called M_raw_. A minimum of 1200 total cells was analyzed for each group. Experimental data are shown as mean ± standard error mean (SEM). As the percentage of dead cells varies significantly between different untreated tumor specimen due to variation inherent to surgery, the percentage of dead cells in the different groups was divided by the corresponding value in the DMSO-treated-PDTSC. Thus, the *y*-axis untitled “Cell death/DMSO” represents arbitrary units corresponding to this ratio.

For Figure 5B, mortality in a given condition is obtained by taking the median value among the different z acquisition and ROI. Only conditions associated with a mortality below 55% were kept for further analysis (four slices removed). As there is a correlation between cell mortality on a given slide treated with the DMSO and the other drugs, we normalized cell mortality values using the following formulae:$$M_{norm}=\left(M_{raw}-M_{DMSO}\right)/f\left(M_{DMSO}\right)$$ where *M_norm_* is the normalized mortality for a given drug, typically between 0 and 1, *M_raw_* is the raw mortality of the given drug, between 0 and 100, *M_DMSO_* is the raw mortality of the *DMSO* for the nearby slice, between 0 and 100, *f*(*M_DMSO_*_)_ is the normalization function, depending on the *DMSO*, which evaluates the maximum amplitude of the drug effect on a given slice. It is given by:$$f\left(M_{DMSO}\right)=\;a-M_{DMSO}\;+\;\left(100\;-\;a\right)*\left(1\;-\exp\left(-M_{DMSO}/b\right)\right),$$ where *a* = 15 and *b* = 10, chosen to fit with the maximal mortality.

The correlation analysis was performed using R version 3.4.3 [57], pairwise. The Spearman’s rank correlation coefficient was used to perform a robust analysis. The figure was generated using the corrplot package version 0.84.

### 4.7. Immunohistochemistry Analysis

Sections (5 µm thick) of formalin-fixed, paraffin embedded tumor tissue samples were dewaxed, rehydrated through graded ethanol and subjected to heat-mediated antigen retrieval in citrate buffer (Antigen Unmasking Solution, Vector Laboratories, Burlingame, CA, USA). Slides were incubated for 10 min in hydrogen peroxide H_2_O_2_ to block endogenous peroxidases and then 30 min in saturation solution (Histostain, Invitrogen) to block nonspecific antibody binding. This was followed by overnight incubation with indicated primary antibodies at 4 °C. After washing, sections were incubated with a suitable biotinylated secondary antibody (Histostain, Invitrogen) for 10 min. Antigen-antibody complexes were visualized by applying a streptavidin-biotin complex (Histostain, Invitrogen) for 10 min followed by NovaRED substrate (Vector Laboratories). Sections were counterstained with hematoxylin to visualize nucleus. Control sections were incubated with pool secondary antibodies without primary antibody.

### 4.8. Statistical and Correlation Analyses

Experimental data are shown as mean ± standard error mean (SEM). Statistical analyses were performed using one-way analysis of variance (ANOVA) with multiple comparisons test (GraphPad Prism 6). A *p*-value of less than 0.05 was considered statistically significant.

## 5. Conclusions

Because the PDTSC strategy maintains the landscape of the original tumor sample, including the stromal and the immune tumor compartment, this approach is a relevant model for individualized testing of drug susceptibility to improve clinical success rates [22,58]. In agreement with the key role played by the immune infiltrate in ccRCC, a phase 3 clinical trial (CheckMate214) showed benefits in term of overall survival and objective response rate using an immunotherapy combination (ipilinumab plus nivolumab) versus sunitinib for intermediate and poor-risk patients with previously untreated advanced renal cell carcinoma [46]. Therefore, PDTSC also warrants further investigations to confirm potential correlations between drug sensitivity responses and the level of tumor vascularization and tumor-infiltrating immune cell populations. Finally, a recent study [59] suggests that further work will eventually make this technique useful for personalized clinical immunotherapy.

## Figures and Tables

**Figure 1 cancers-12-00232-f001:**
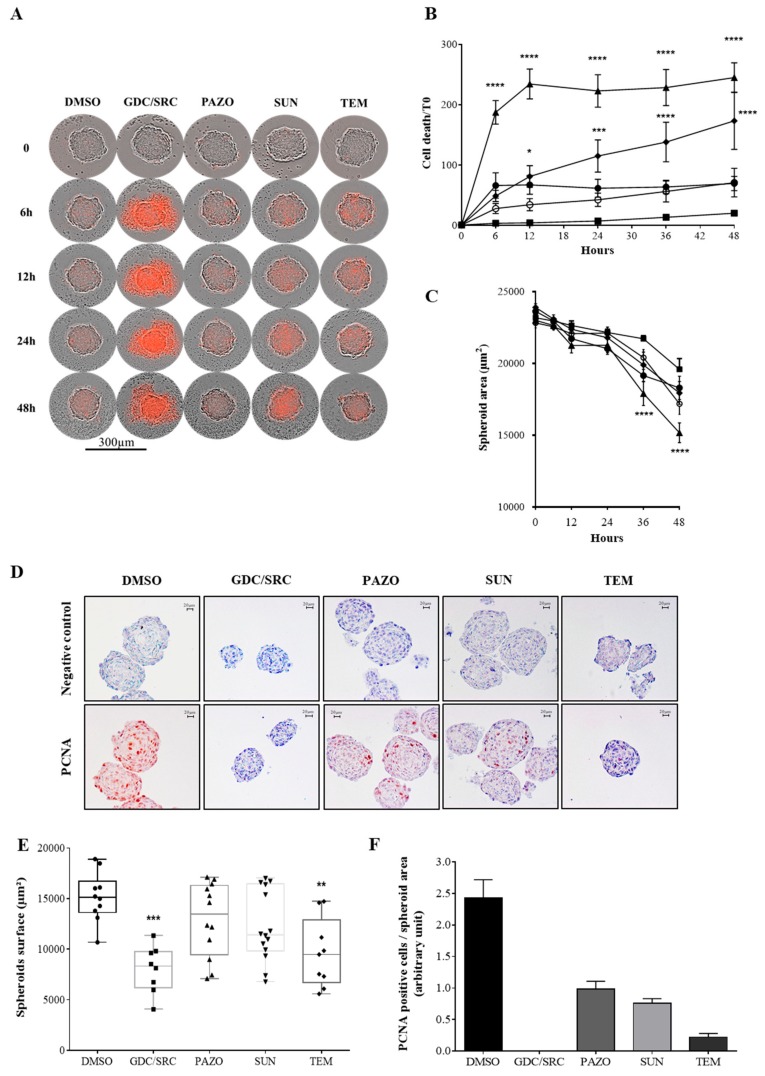
Treatment of 786-O spheroids. 786-O-WT (VHL^-^) cells were grown as spheroids and treated with 10 µM of either GDC-0941 + saracatinib (10 µM each, GDC/SRC, ▲), pazopanib (PAZO, **○**), sunitinib (SUN, ◆), temsirolimus (TEM, ●), or vehicle (DMSO, ■) in the presence of propidium iodide. Cell death was monitored on spheroids using either an Essen IncuCyte Zoom live-cell microscopy incubator or by immunohistochemistry. (**A**) Bright field and fluorescent overlaid images show 786-O-treated spheroids at indicated times (0, 6, 12, 24, and 48 h). Bar scale 300 µm. (**B**) Images taken automatically every 6 h over 48 h of culture were analyzed for PI fluorescent area quantification. Cell death values (PI labeling area) was divided by the corresponding spheroid area and multiplied by 100. This percentage of cell death was divided by the one at T_0_, for all the others time points and was expressed as mean ± SEM. The statistical analysis of dead cells was performed with 2 way ANOVA test for each time point compared to DMSO treatment. (**C**) The same images were analyzed for spheroid area quantification. Significant difference was observed between GDC/SRC (**** *p* ≤ 0.001), SUN (**** *p* ≤ 0.01), TEM (**** *p* ≤ 0.01) versus DMSO after 36 h of treatment using a Kruskal-Wallis test. (**D**) PCNA staining to visualize proliferation of fixed paraffin-embedded (scale bar, 20 µm). (**E**) Spheroid area quantification by surface calculation of (**D**), (*n* ≥ 8). Significant difference was observed between GDC/SRC (*** *p* ≤ 0.001), TEM (** *p* ≤ 0.01) versus DMSO in a Kruskal-Wallis test. (**F**) The number of PCNA positive cells was quantified in each spheroid and divided by the corresponding spheroid surface. Histogram plot represents mean of PCNA-stained cells pooled from 4 to 6 spheroids (biological replicated/condition) with error bar (±SEM).

**Figure 2 cancers-12-00232-f002:**
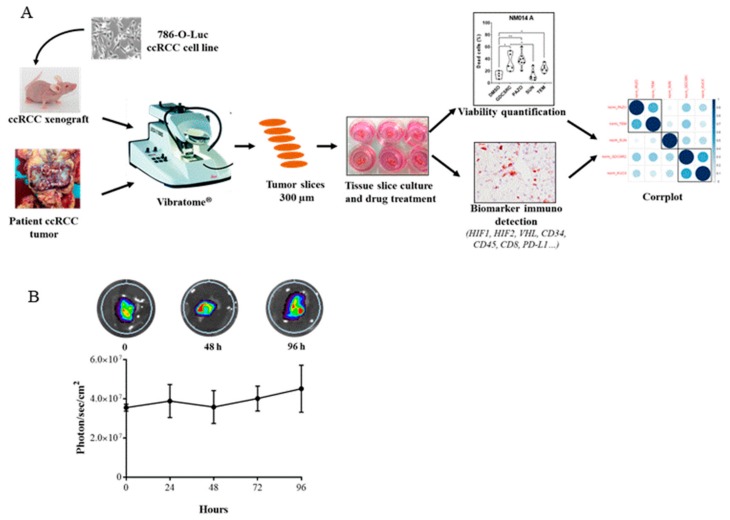
The procedure flowchart for renal tumor slice culture. (**A**) 786-O-derived tumors generated in mouse xenografts or human ccRCC surgical resection specimens are cut into 300 µm slices in buffer solution using a Vibratome^®^. The slices are transferred to culture medium and then carefully placed on membrane insert in 6-well plates to create an air-liquid interface. After 48 h of drug treatments, slices are analyzed for cell viability and biomarker immuno-detection. Correlation between drug sensitivity and biomarker expression is visualized with the graphical display of a correlation matrix (Corrplot, R package). (**B**) Tumor slices maintain cell survival over four days of culture. Slices from 786-O-luc-derived tumors were cultured for up to four days, with fresh media changes performed every two days. Each day, luminescence was recorded from slices after luciferin addition using IVIS imaging (upper panel). Plotted normalized photon quantification showed minimal changes over the culture periods.

**Figure 3 cancers-12-00232-f003:**
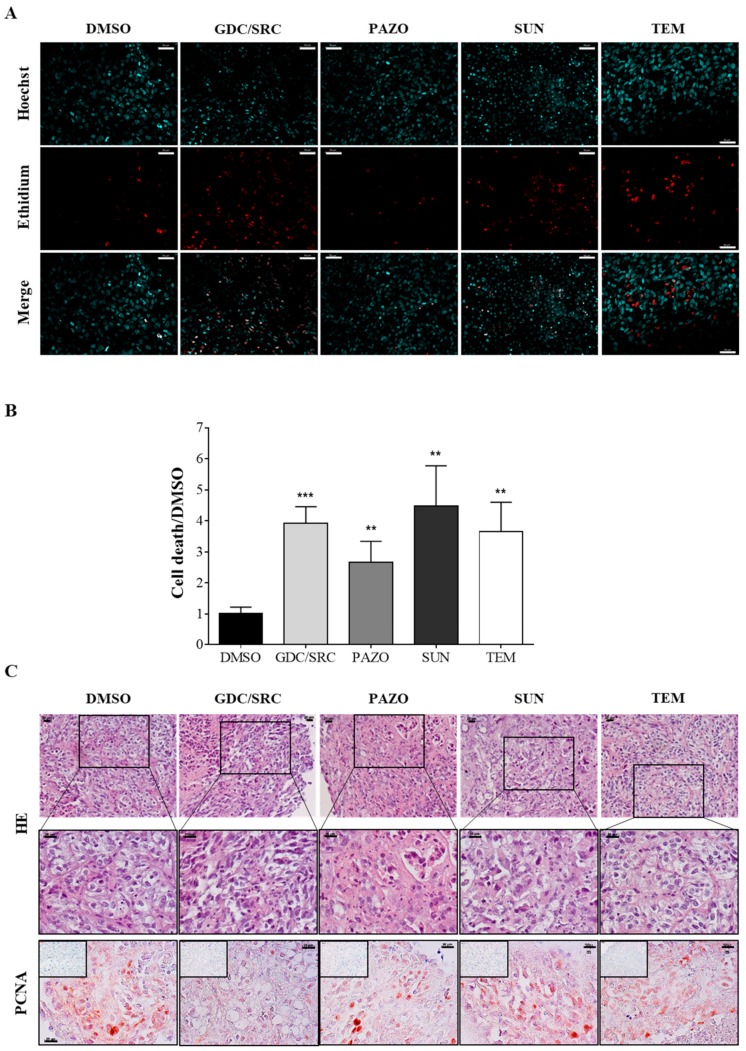
Treatment of slice cultures from 786-O tumor xenografts. 786-O cells were injected under the renal capsula of Balb/c nude mice. One month later, mice were euthanized, tumors were harvested and processed for tissue slice cultures. (**A**) Tissue slice cultures were treated with 10 µM of either GDC-0941 + saracatinib (10 µM each, GDC/SRC), pazopanib (PAZO), sunitinib (SUN), temsirolimus (TEM), or vehicle (DMSO 0.2%) for 48 h. Nuclei were stained with Hoechst 33342 and dead cells were visualized by Ethidium homodimer staining. Images were taken with an Apotome-equipped Zeiss microscope. Bar scale 50 µm. (**B**) The intensity of Ethidium homodimer positive cells was measured in each nucleus on five independent areas of the tumor slices as described in Material and Methods. The *y*-axis represents the ratio of the percentage of dead cells in the different groups divided by the corresponding value in the DMSO-treated-slices. Significant differences in cell death were observed between DMSO versus the GDC/SRC combination (*** *p* ≤ 0.001) or each drug alone (** *p* ≤ 0.05) using a Mann–Whitney test. (**C**) Tumor slices were treated as described in A, then fixed and embedded in paraffin. Fixed tissue slices were stained with Hematoxylin-Eosin (HE). Representative pictures of treated slices are shown at two magnifications (lower magnifications, upper images and higher magnifications, middle images). Tumor slices were also stained with the anti-PCNA antibody to visualize cell proliferation (lower panel). Negative controls (no primary antibody) are shown in the insets. Scale bars 20 µm.

**Figure 4 cancers-12-00232-f004:**
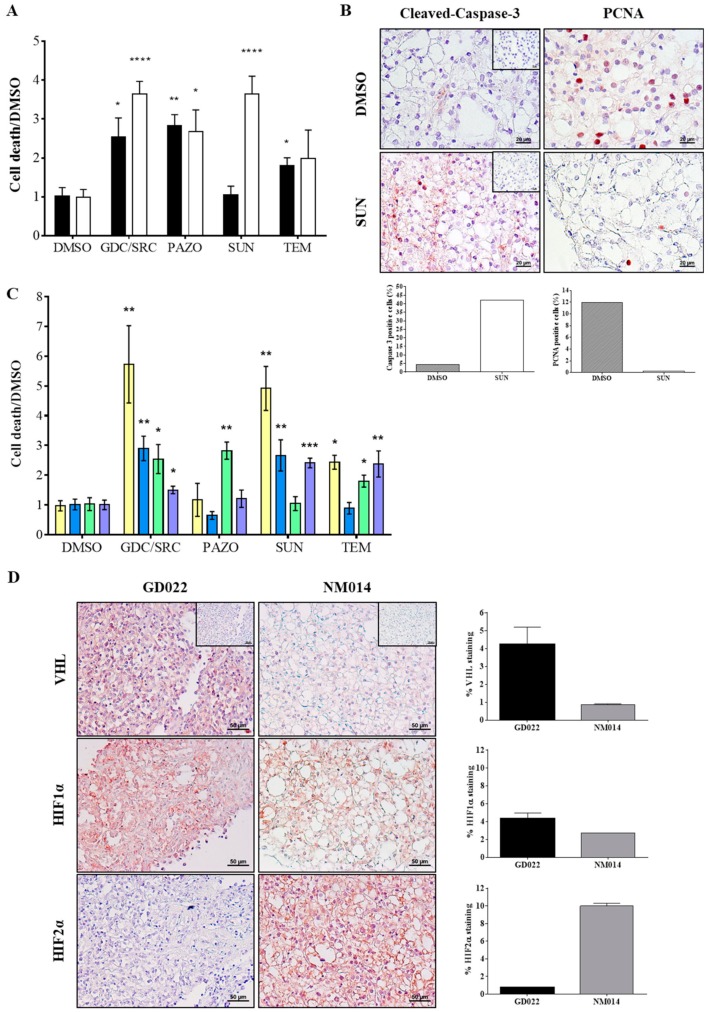
Treatment of slice cultures from human renal tumors. Tissue slice cultures from human renal tumors were treated for 48 h with a panel of drugs (10 µM each) and cell viability assayed as in Figure 3A. (**A**) Intra-tumor heterogeneity. Two fragments A and B of the same tumor (NM014) were analyzed for their sensitivity to indicated drug treatments. Mean DMSO was normalized to 1 to compare the two fragments of NM014. The *y*-axis represents the ratio of the percentage of dead cells in the different groups divided by the corresponding value in the DMSO-treated-PDTSC. Cell death measurement in fragment A (black bars) from NM014 shows significant differences between DMSO versus the combination (GDC/SRC, * *p* < 0.5), pazopanib (PAZO, ** *p* < 0.01) and temsirolimus (TEM, * *p* < 0.5) but not versus sunitinib (SUN). The same analysis of fragment B (white bars) from the same NM014 tumor, shows similar profile except for sunitinib that in this case induced significant cell death (SUN, **** *p* < 0.0005). (**B**) Apoptosis and proliferation assays. Representative pictures of tumor slices from fragment B of NM014 treated for 48 h with DMSO (**upper panels**) or 10 µM sunitinib (SUN, **lower panels**) and stained with Cleaved-Caspase-3 (**left panel**) or with anti-PCNA antibody (**right panel**). The PCNA stain identifies cells that are proliferating while the Cleaved-Caspase-3 stain shows cells undergoing apoptosis. The percentages of PCNA and Cleaved-Caspase 3 positive cells were plotted below each set of pictures. Scale bars, 20 µm. Negative controls (no primary antibody) are shown in insets. (**C**) Inter-tumor heterogeneity. Four different tumors were treated and analyzed as in Figure 3A showing distinct drug sensitivity profiles. Each color represents one patient tumor (Yellow, NB029; Blue, YL024; Green, NM014; Purple, MD034). (**D**) VHL and HIF expressions. Representative pictures of two untreated tumor slices GD022 and NM014 stained with anti-VHL, anti-HIF1α or anti-HIF2α antibodies. Scale bars, 50 µm. For each staining, images taken from five independent areas of a tumor slice were quantified with ImageJ and plotted as percentage of specific staining relative to tumor area (respective right panels).

**Figure 5 cancers-12-00232-f005:**
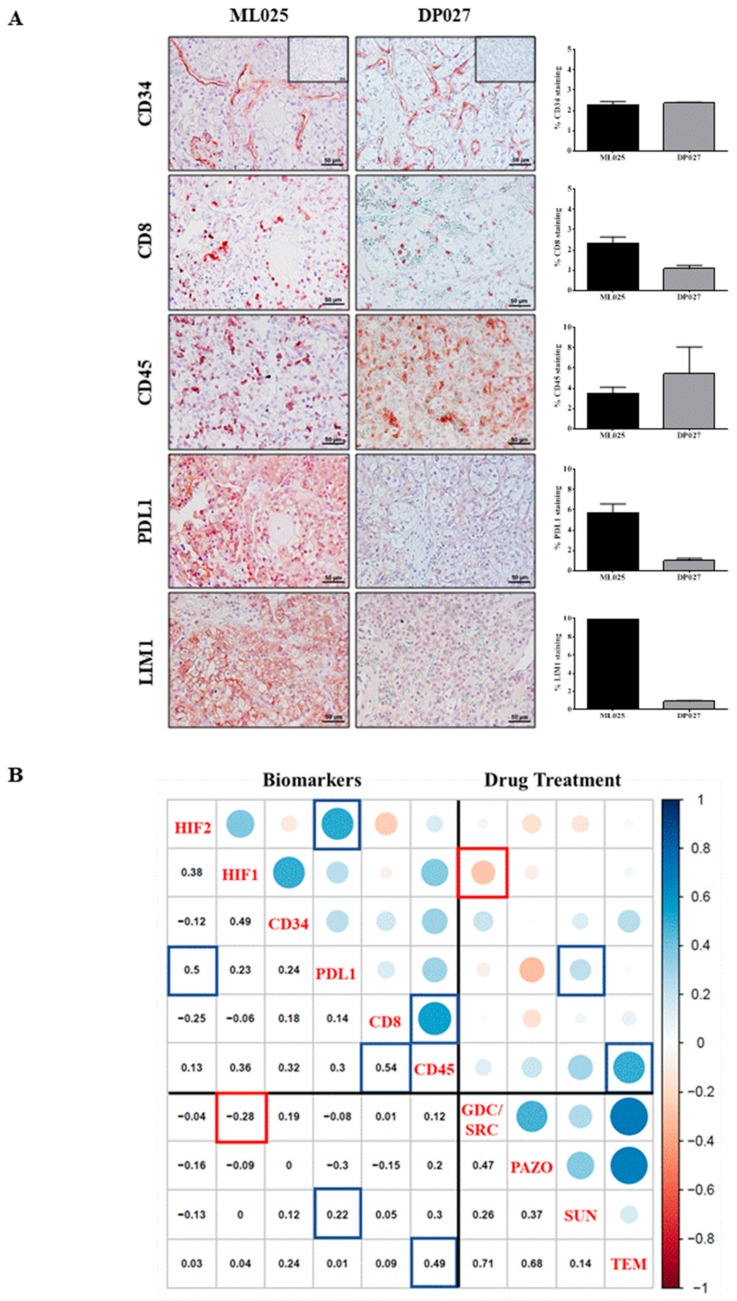
Predictive biomarkers in renal tumor slice cultures. (**A**) Vascular, immune and stem cell type characterization. Representative pictures of untreated tumor slices ML025 and DP027 stained with the following antibodies: anti-CD34, anti-CD8, anti-CD45, anti-PDL1, anti-LIM1. Scale bars, 50 µm. For each staining, images taken from five independent areas of a tumor slice were quantified with ImageJ and plotted (**right panels**). (**B**) Correlation plot between the percentage of positive cells following various IHC staining and the normalized proportion of dying cells following application of drug treatments. The Spearman rank correlation was used. The diagonal indicates the biomarker used for the IHC staining (**left part**) or the drug treatment (**right part**). Below the diagonal is the pairwise correlation value, and above the diagonal is the corresponding representation, with the color legend that is the bar on the right side of the plot. Blue (resp. red) colors correspond to positive (resp. negative) correlations. Boxes correspond to cases described in the text. For example, the two blue boxes on the top-left side highlight a correlation of 0.5 of the percentage of positive cells between HIF2 and PDL1 staining.

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
