# Peer review of "Ex-Vivo Treatment of Tumor Tissue Slices as a Predictive Preclinical Method to Evaluate Targeted Therapies for Patients with Renal Carcinoma"

_cancers, 2020, doi:10.3390/cancers12010232_

Round 1

Reviewer 1 Report

Roelants et al., report methods for testing human tumor samples cultured ex vivo from patients diagnosed with the disease clear cell renal cell carcinoma (ccRCC). These methods have translational applications for assessing an individual patient’s potential response to therapeutics and new drug screens. They developed three explicit models: (1) 3D tumor spheroids; (2) tumor xenographs in mice; (3) patient-derived tumor slice cultures (PDTSC). The authors tested sensitivity to a battery of assorted drugs, and used read-outs such as cell death assessed using staining for propidium iodide followed by quantification with the Incucyte microscopy instrument. The results of their PDTSC studies are very interesting. The authors provide here a proof-of-concept that analysis of patient samples can be utilized to study cellular traits of tumors from ccRCC.

Points to address:

1. Figure 1: Why does the spheroid area decline in DMSO treated spheroids? Regarding Panel B: What unit is on the y-axis? Please explain. Panel D: Please provide a quantification for the number of cells that show PCNA staining in the control and experimental groups.

2. Figure 3: Rather than reporting relative intensity of staining, cell number should be quantified and reported.

3. Figure 4: Panel A: What unit is on the y-axis? Please explain. Panel B: Provide absolute counts of cells for each stain and control versus experimental groups.

Reviewer 2 Report

In this manuscript the investigators used thick tissues slices to study whether these could be informative as companion tool to be used to stratify metastasized RCC patients for which multiple first line treatments are available. As correctly stated by the investigators, the use of patient-derived xenografts and cancer-organoid models for this purpose may be limited in view of the time needed to establish these while the decisions for therapy choice need to be made much more rapidly. The manuscript is well written and has a clear message.

Comments:

The authors screened primary RCC, which is understandable. They should emphasize that fresh primary tissue may not be available when needed (in case of recurrent disease). Additionally, radical nephrectomy is not always performed on metastasized patients, whereas screening of these patients would be highly beneficial (provided that the assay is predictive). Indeed, this is also true for the PDX and organoid model, but in view of their statement that this may become a diagnostic companion tool, they should discuss these issues as possible difficulties that need to be overcome before implementation would be feasible.

Some of the drugs tested are anti-angiogenic drugs. Nevertheless, the read out was on the ccRCC cells. It would be wise to comment on this fact. In the figures the effects on the tumor vascular appear to be minimal, which is surprising. An explanation for this observation is warranted.

The concentrations of the drugs used is not mentioned anywhere. Without this information it is impossible to know whether the drug concentrations are similar to those reached in patients, or much higher (in which case the relevance may become questionable)

The active drugs are in some cases drug metabolites (e.g. for sunitinib). This cannot be mimicked in the thick slices. The effects for these drugs are therefore difficult, if not impossible to interpret: the effects may be completely irrelevant. (unless the active metabolites were tested, but this cannot be assessed from the manuscript).

Round 2

Reviewer 2 Report

The authors addressed my concern adequately.